# Feasibility Study of the Flatness of a Plastic Injection Molded Pallet by a Newly Proposed Sequential Valve Gate System

**DOI:** 10.3390/polym14030616

**Published:** 2022-02-04

**Authors:** Hsi Hsun Tsai, Yi Lin Liao

**Affiliations:** 1Department of Mechanical Engineering, Ming Chi University of Technology, New Taipei City 24301, Taiwan; M09118018@mail2.mcut.edu.tw; 2Research Center for Intelligent Medical Devices, Ming Chi University of Technology, New Taipei City 24301, Taiwan

**Keywords:** plastic pallet, injection molding, flatness, sequential valve gate system, molding flow analysis

## Abstract

The investigation of plastic pallet molding, assisted by a sequential valve gate system, has not yet been performed due to the limitations of the pallet scale. Furthermore, at present, the application of recycled plastics by chemical industries has become extremely popular around the world. This study aimed to determine pallet flatness experimentally and numerically using recycled polypropylene with a large-scale pallet. Short-shot testing on injection molding was performed to obtain short-shot samples for confirmation of the flow front during simulated filling. The real injected pallet profile, which was measured by an ATOS, was compared after confirmation to the numerical profile of the pallet. The pallet’s flatness was accurately compared to the real experimental and numerical results. By adjusting the temperature of the cooling channel within the cavity plate to 55 °C, the flatness of the pallet achieved by the newly proposed sequential valve gate-opening scheme was about 7 mm, which meets the height directional warpage standard determined by the pre-set sequential scheme. The numerical flatness is in line with existing flatness values for pallets. Furthermore, the proposed cooling temperature gives the highest yield in terms of pallet molding from the perspective of the stakeholders.

## 1. Introduction

As defined in the SFS-EN ISO 445 standard, a pallet is a “rigid horizontal platform of minimum height, compatible with being handled by pallet trucks, forklift trucks and/or other appropriate handling equipment” and can be used “as a base for assembling, loading, storing, handling, stacking, transporting, or displaying goods and loads” [1]. Pallets can be made of wood, plastic, aluminum and composites, and used under the three pallet management strategies of single use, buy/sell, and pooled. A pooled pallet is leased to customers without transfer of ownership. A standardized pallet is designed to last several trips, under a scheme called the “buy/sell” strategy. However, single-use pallets are the simplest strategy, as they are discarded after one trip [2], after having been loaded with goods and transported by container ships to places all over the world. Recycled plastic pallets were found to be superior to conventional plastic pallets by an impact category analysis of the results per trip. The recycled plastic pallets also performed better in terms of environmental impact compared of wooden pallets [3]. Single-use pallets made of recycled plastic via injection molding are the subject of this study.

Recycled plastics are based on recycled PE (rPE) and recycled PP (rPP), and they have outstanding potential. They have the potential to significantly contribute to new markets with more demanding and critical applications. One of the applications with the highest demand is found in the lower-quality end of the agricultural and building sectors, serving as a structural part [4]. Gall et al. [5] revealed the properties of the recycled plastics, and found that the density, melt flow rate (MFR) and Charpy impact strength of the rPP materials varied from 0.904 to 0.924 g/cm^3^ (by ISO 1183), from 13 to 22 g/10 min (by ISO 1133), and from 5.9 to 6.8 kJ/m^2^ (by ISO 179). The content of calcium carbonate in the rPP, analyzed by a thermos-gravimetric analyzer (TGA), ranged from 0.29 to 1.92% in mass. As they contain diverse contaminants, recyclables should be blended with legacy calcium carbonate and a polymeric cross-contaminant to modify the MFR for later applications in plastic engineering processes. One of the more common industrial plastic-processing technologies is plastic injection molding.

When using rPP in micro-injection molding, a uniaxial extension test showed that the increases in Young’s modulus, yield stress and ultimate stress values were 3.07%, 10.97% and 27.33%, respectively. A 1.29% reduction was found in the breakage strain compared to virgin PP samples [6]. The variations in the recycled plastic’s properties may disturb the final quality of the injection part, due to the injection parameter setting remaining constant throughout the whole process. An important quality of the injection part is the warpage, which can face problems related to a combination of poor material characterization and inadequate control of the processing parameters [7,8,9,10,11]. The temperature differences between the two mold surfaces significantly affect the morphology distributions of the molded parts. The cooling rate may affect the injection parts in terms of relaxation/reorganization levels and give rise to an asymmetric distribution of mechanical properties [12]. An imbalance in the mold-filling is one of the factors affecting the asymmetric temperature distribution of the injection part [13,14]. Plastic injection molding involves four major stages: filling, packing, cooling, and ejection. The injection pressure and rate, packing pressure and time, and cooling temperature and time may affect the quality of the injection part. The mold temperature seems to be one of the main process parameters that affects the properties of molded parts [15,16]. Nevertheless, the mold temperature is actually unstable during the process. The temperature of the mold has never been controlled individually, as it is affected by the cooling channel and cooling time, as well as the mold opening time.

Filling the mold cavity with melted material via a gate is essential for the small injection part. Considering the limitation of the flow length from the gate, multi-gate filling can reduce a machine’s required injection parameters, and the filling time, during injection molding. Not using multi-gate filling injection is associated with a higher number of weld lines. A weld line is formed when two separate melt fronts join into one flow. Moreover, while weld lines are not appropriate for parts, it is impossible to avoid all of them due to the filling efficiency of large molding components. It is well known that the strength of weld lines is lower than the strength of the general plastic-molding material, since a lower temperature, along with air bubbles, occurs between the fronts. The structural parts molded by rPP, such as the pallet and water cage, endure the external load. The weld lines within the molded structure parts face an increased risk due to the injection molding of plastic pallets via multiple gates, whereby the melted materials flow into the mold cavity from a molding machine.

The concurrent filling of multiple gates increases the filling pressure, meaning that a larger injection molding machine is required to clamp and pack the mold during the filling and packing processes. A sequential valve gate-opening system can be used to decrease the clamping and packing forces, which divides the gates into several groups during the filling stage. Via this sequential valve gate-opening system, the flow front from the initial gates spreads to the lateral gates. The lateral gates are activated to pass on the melt material when they come into contact with the front. This approach could eliminate many weld lines from the molded part. Moreover, the scale of the injection machine can be decreased. During the injection molding of ASTM-D638 standard specimens (200 mm in length, 20 mm in width, and 2 mm in thickness) made of acrylonitrile butadiene styrene (ABS), sequentially setting the on/off times of these filling gates enabled the sequential valve gate system to eliminate the welding lines and increase the tensile strength [17]. In addition, the different temperature levels of the melted material’s flow fronts tensile strengths, and melt polyamide flow front temperature (PA6) were correlated with the strength of the welding line [18].

The sequential valve gate-opening system could be implemented to create large-scale plastic parts with a moderately sized injection machine. Knowing the flow front of the melt material in the mold cavity is essential for setting the switching time of all the gates in the sequential valve gate system—a theoretical model or a numerical approach could be used to predict the flow front of the injection in the mold cavity. Iwko et al. [19] derived numerical results to verify the experimental results by constructing a comprehensive model of the plasticization process in a screw-barrel system injection molding machine. They found that the output pressure and temperature of the plasticization process, determined numerically by the model, fit the experimental results with an average error of less than 10%, but the flow front in the mold cavity was never assessed.

Cardozo [20] reviewed the numerical approaches to filling via injection molding, and indicated that the Moldex3D software, a commercial software available for injection molding, could provide an understanding of the physical effects occurring in the mold cavity. Moldex3D was applied to investigate the molding process, while the prediction of the flow front during filling was derived from the Hele–Shaw model [21]. The finite-volume approach was used to determine the multi-physical quantities involved in the packing and cooling processes. Furthermore, the equation governing the jetting behavior of the filling from the gate was discretized by a control volume-based, finite-volume method [22]. By using each of the commercialized software, including ANSYS Fluent [23,24], Moldflow [25], Open FOAM [26] and Moldex3D [10,27,28,29,30], one could analyze the multiple physical parameters of the output molding parts in relation to the operational parameters of the injection molding process. Notably, the experimental investigator could easily make a comparison between the simulation results and the real operational results in the laboratory.

Using Moldex3D, the authors undertook a numerical feasibility study of a single-use pallet created via a sequential valve gate system [10]. However, the experimental results of this rib-structured pallet have not been determined. Differing from previous rib-structured pallets, the target pallet has a flat top surface. The aim of this study is to investigate the temperature, pressure, stress and warpage of the injection-molded rPP flat surface pallet with size dimensions of 1 m × 1 m × 0.13 m by CAE simulation and experimental methods. The results for the numerical warpage of a plastic pallet produced via a sequential valve gate system, derived by Moldex3D 2020, are compared with the real profile of the rPP pallet measured by the ATOS scan box 5120 system. Using the injection parameters of the plastic pallet, the specifications of the injection machine, and a polymer database, a fundamental investigation can be conducted to understand the injection molding process. By using a sequential valve gate-opening system to mold a plastic pallet, the flow fronts during the filling stage can be derived for an evaluation of the actuating time of the gates. A detailed understanding of the pre-setting sequence applied to the valve gates may help to determine the ideal sequence. A pre-set valve gate-opening sequence, provided by the cooperating company, would enable a comparison of the flatness of the pallet. A new proposed valve gate-opening sequence can then be used to derive an improved pallet flatness, which will help to evaluate the accuracy of the numerical predictions.

## 2. Experimental Setup and Software

Figure 1 shows the dimensions of the plastic pallet that were used in this study. It has a flat top surface and a reinforced structure with ribs at the base. The ribs are 3 mm in thickness and were changed according to the draft angle used for the injection molding. A general rule when arranging the positions of the filling gates is to ensure that the ratio of flow length to rib thickness is below 150. When the ratio is lower than 150, the running length of the melted polymer within the mold during injection is sufficient to maintain the melting state. In this application, the maximum filling length from gates #1, #4, #13 and #16 to the four corner legs of the pallet was 394 mm. The ratio of the flow length to rib thickness in this study was 131.3, which is smaller than the general standard when arranging the positions of the filling gates. As such, there were 16 filling gates (7.0 mm in diameter and 20 mm in height), as shown in Figure 1a. rPP was used as the injection molding plastic material to simulate real-world plastic waste reduction. The pallet’s dimensions were 1000 mm in length and width and 130 mm in height, as shown in Figure 1b. An isometric view of the pallet’s base (Figure 1c) shows the complex features that should be cooled during the molding process by the baffle cooling flow system, as shown in Figure 1d. The molding experiments included sequential gate-opening and concurrent valve gate-opening systems for the melt-filling process, in order to numerically analyze the mold flow. The gates’ opening and closing were sequentially controlled by a pneumatic system, so that the previous gates were closed by relay during melt filling, and the following gates were concurrently opened—this assumes that the melted material starts to fill the mold cavity once the valve gate is completely open.

Within the simulation, the 3600-ton injection, performed by the Supermaster 3600E1 molding machine (https://chenhsong.com/, Taoyuan Taiwan, 3 February 2022), is modeled. This molding machine has a screw diameter of 225 mm, a maximum screw stroke of 4400 mm, a maximum injection pressure of 159.7 MPa, and a maximum injection volume of 49,278 cm^3^. The simulation analysis was performed using Moldex3D software. In this software, both the skin and core materials are considered to be compressible, generalized Newtonian fluids. The surface tension at the melt front is neglected. The modified Cross model with Arrhenius temperature dependence was employed to describe the viscosity of the polymer melt. During the polymer melt-filling phase, the velocity and temperature were specified at the mold inlet. While the core material was injected, the flow rate was specified at the mold inlet. On the mold wall, the non-slip boundary condition was applied, and a fixed mold wall temperature was assumed.

In Moldex3D, the finite volume method was used to discretize the Navier–Stokes equation based on the pressure-based decoupled procedure and solve the transient flow field in a complex three-dimensional geometry. A compressive, bounded, high-order differencing scheme was also utilized to directly solve the hyperbolic advection equation of the fractional volume function to track the melt front during the filling process [21]. Modeling the flow field in Moldex3D is an iterative decoupled procedure for coupling velocity and pressure, in which the three linearized momentum equations are solved for an estimated pressure field, then sequentially followed by the solution of the pressure correction equation. The mass fluxes and pressure are then corrected. This will satisfy both local and global continuity, but can cause the momentum to deviate. Hence, a new outer iteration is activated. The process is then repeated until the prescribed tolerance for each equation is achieved [21].

All sixteen of the valve gates are concurrently opened to allow the molten material to enter, with a hot runner used to determine the filling flow front, the clamping force, the temperature distribution, the thermal stress, and the deformation; then, the 16 gates are opened and closed in a controlled sequence. The pre-set sequence of the sequential gate-opening scheme is depicted in Table 1. Gates 3, 5, 9, and 13–16 were initially opened to fill the mold cavity with the melted material. Gates 8 and 11 were then actuated within the first second. After this filling, gates 1, 10, and 12 were relayed. Gate 4 was turned on in the third second, and then gate 7 was activated after five seconds. Gates 6 and 2 were opened in the sixth and eighth seconds, respectively, until the end of filling. The total filling time was 9.3 s.

Using the previous setting sequence, the Moldex 3D 2020 software was used to simulate the sequential valve gate system used for the melt-filling process in the mold flow analysis. During melt-filling, the gate-opening and -closing times are sequentially controlled by the pneumatic system, so that the previous gates are closed by relay, with subsequent gates opening at the same time. This software package was also used to simultaneously open all 16 gates and direct the melted material into the mold cavity, which is the concurrent valve gate-opening scheme. The distribution of the weld lines, the filling pressure, and the estimated clamping force can also be derived. The molding analysis was then conducted for the same injection filling time via the sequential gate-opening scheme. The molding pressure within the mold cavity was measured numerically, using the same injection parameters used to set the timing control of the 16 gates. Under the sequential gate-opening scheme, the hot runners were opened at different times. Within the pallet, the temperature distributions, filling pressures, deformations and thermal stresses, as well as the shrinkages, were compared under the sequential gate-opening and concurrent valve gate-opening schemes.

During the filling stage, the fill flow front of the molten material is closely dependent on the viscosity, the material temperature, and the runner and gate of the mold. The pressure of the molten material is usually a consequence of viscosity. During the filling stage of the injection molding process, the flow front is controlled by the flow rate and pressure gradient. Adding mica powder to the rPP gives the material properties [10] such as those shown in Table 2, with the density of rPP being 1.026 g/cm^3^, which is more than that of raw PP. In the design phase, the volumes of the plastic pallet and mold were 10,039 and 1,179.92 cm^3^, respectively. The associated solid mesh of the plastic pallet, mold, and cooling channel contained 1,562,598, 4,724,368, and 4,189,448 elements, respectively. The volume and the mesh number of the 16 hot runners were 784.06 cm^3^ and 839,450 elements, respectively. The surface mesh of the pallet contained 311,368 elements. The experimental viscosity with respect to the shear rate and temperature, the specific heat with respect to temperature, and the mechanical properties of the melt rPP resin were derived from Cheng [10]. The injection molding parameters of the rPP that were used for this plastic pallet are shown in Table 1.

It was assumed that the melted material would begin filling the mold cavity once the valve gate was completely open, and the 16 gates were opened sequentially to direct the melted material into the mold cavity. The flow fronts that occurred during short-shot testing in the injection molding of rPP pallets were compared to ensure correct simulation modeling. Then, the most suitable sequential control scheme of the gates with the same injection parameters was investigated. Via the appropriate sequential control of the gates, the filling pressures, temperature distribution, warpage deformation, thermal stresses and shrinkages were assessed to determine the advantages of this analysis.

The rPP pallet molded by the Supermaster 3600E1 molding machine is 1 m in length and width and 0.13 m in height. This pallet scale is too large to measure its three-dimensional profiles using a general coordinate measuring machine (CMM). The ATOS scan box 5120 system (GOM, Swiss, www.gom.com, 3 February 2022) is a non-contact three-dimensional measuring system that operates via high-speed sensors used to scan all the parts. A 3D graph could be generated from these scans for 3D printing, reverse engineering or part inspection. The injection-molded pallet was measured by RATC in Taiwan (https://www.ratc.com.tw/, 3 February 2022) with ATOS scan box 5120. The authors imported the 3D graph generated by the ATOS scanning system into Creo Parametric CAD to calculate the flatness of each surface on the pallet. The measured flatness and the profile of the pallet’s surface were compared with the numerical results derived by Moldex3D.

## 3. Results and Discussion

### 3.1. Flow Front Comparison of Short Shot Testing

When producing a fully molded part via the plastic injection molding process, the screw-back position refers to the screw within the injection machine barrel being brought back to the starting position before the start of the next cycle. Incomplete filling of a part, called short-shot molding, can be achieved in plastic injection molding by manually shortening the screw-back position to check the real flow fronts that propagated from the filling gates during the filling stage. The implementation of short-shot testing may help investigators to estimate the propagating times of the initial flow fronts from the filling gates to the next gate. Using previously recorded times, one could propose an actuating sequence for the sequential valve gate-opening system. In addition, the real shapes of the short-shot molding samples and the simulated flow fronts can be verified to complete the numerical validation.

The concurrent opening of all the filling gates produces a higher injection pressure and, thus, a higher clamping force in general plastic injection molding. The concurrent opening scheme is different from the sequential valve gate-opening scheme. Short-shot simulations of 30% and 60% filling are shown in Figure 2. Figure 2a,c show the flow fronts when the 16 gates are opened at the same time. The fill flow fronts that form around gates 2 and 3, 6 and 7, 10 and 11, and 14 and 15 initially interact, and are stitched at the half center of the plastic pallet. The fronts are expanded to the four corners, which then become the portions with the longest flow lengths during injection mold filling. The color distribution is almost uniform, since all gates are open. Weld lines are located on the borders of each of the pairs of gates, and there are 17 borders on the top surface of the pallet. The other borders are located across the pallet.

In terms of weld lines, variant filling can be improved by the one-gate or sequential valve gate-opening schemes. In this study, for a large-scale plastic pallet, a pre-set sequence of opening the valve gates was used to expand the filling fronts. The pre-set sequence was provided by the industrial company we are working with. Figure 2b,d show that the 16 gates are controlled in a seven-stage sequence, as also shown in Table 2. The comparison of the filling fronts produced under the concurrent opening and sequential valve gate-opening schemes is shown in the form of iso-surfaces. An iso-surface is a surface produced over an equal time within a volume of space. Changes to the iso-surfaces’ shapes show differences close to the gates—that is, the flow fronts of the melt are increased close to these gates. The iso-surface shows a higher velocity of melt material in the middle of the pallet. During the sequential valve gate-opening scheme, the gates are not ideal due to the number of weld lines, while this is slightly decreased under the concurrent opening scheme. Furthermore, the initial gates that were opened under this sequential pre-set valve gate-opening scheme are inappropriate for real-world applications.

The real temperatures of the core and cavity plates of the mold were unknown before the filling stage of the injection process. The mold temperature was increased with a shortened cycle time. As the residual heat within the mold, which would not be transferred into the cooling channel, increased, the mold temperature is unknown, although the mold temperature can be detected at the ejection stage. However, the detected mold temperature cooled down during the ejection stage. A preliminary analysis of the flatness can be performed by assuming the mold temperature. A further serial numerical analysis would help us to determine the exact mold temperature by adjusting the cooling channel mold temperature to fit the experimental surface flatness of the pallet. By pre-setting the sequential valve gate-opening scheme (Table 2), a serial short-shot test can be performed experimentally. A 70% short-shot sample was produced by the Supermaster injection machine (Figure 3a,c), showing the top and bottom surfaces of the rPP injected pallets). The 70% short-shot of the real injected pallet was compared to the fronts on the top and bottom sides of the pallet via a numerical approach (Figure 3b,d). The profile of the real pallet qualitatively agreed with the simulated flow front. In Figure 3e, the profile of the top surface of the pallet produced by 90% short-shot testing indicates a close fit with the numerical flow front, as shown in Figure 3f. The bottom side of the 90% short-shot pallet provides significant evidence of the prediction accuracy of the experimental and numerical results shown in Figure 3g,h. Figure 3b,f show that the imbalanced flow fronts spread to the entire pallet due to the inaccurate setting of the valve gate-opening sequence. This mis-setting may induce a fluctuation in the spread front. Through the simulation results, a suitable sequence of the valve gates’ switching times can be predicted. The flow front thus spreads continuously once it meets the next gate.

### 3.2. Flatness

The temperature of the cooling channel within the mold plate, including the core and cavity plates, was set to 20 °C. The injection molding samples were acquired after 20 cycles to ensure that the temperature of the mold plate was in a steady state. Then, the temperatures of the mold’s core and cavity were measured by K type thermocouple after the pallet was ejected from the core plate. These two measured temperatures were higher than the cooling channel temperature since their heat is never effectively transferred to the cooling channel. The exact temperatures of the mold’s core and cavity could not be measured during the packing and cooling stages of injection molding. To fit the numerical warpage to the molded one using the trial-and-error method, an empirical approach was used to adjust the temperatures of the cooling channels within the cavity and core plates of the mold, which induce different temperature levels in the two plates compared to the predicted temperature for the core and cavity of the mold. Hence, the entire pallet was measured by way of the ATOS scan box 5120 system, and this was then compared with the numerical results.

The difference between the highest and lowest positions of a surface profile in the height direction can be called the flatness. Figure 4a depicts the measured flatness of the top surface of the molded pallet by the previous ATOS system. The entire profile of the pallet’s top surface is convex and has an excellent flatness value of 6.721 mm. By adjusting the cooling channel temperature of the mold’s cavity plate to 55 °C, and the core plate temperature to 20 °C, for simulation on Moldex3D, the pallet’s numerical top profile is symmetrically convex and the surface flatness of this top surface is 6.044 mm, as shown in Figure 4b. This molded pallet has nine legs to support the entire top surface and the loads. The flatness of the pallet’s nine-leg plane is, therefore, important for stably supporting goods. The flatness of the nine-leg plane of the pallet’s bottom surface is acquired by an ATOS of 6.772 mm, as shown in Figure 4c. The predicted flatness of the bottom side of the pallet is 5.960 mm in Figure 4d. The deviation of 0.812 mm, with regard to the 130 mm pallet height, is relatively accurate. 

We compared the results, showing the true measured pallet height in the ideal coordinate system by ATOS with the reconstructed deformations in Figure 4a and the numerical heights produced by Moldex3D in Figure 4b. A comparison of experimental and numerical profiles of the molded pallet along the x axis is shown in Figure 4e, where the purple crosses denote the heights measured every 50 mm along the middle (*x_c_*) of the top profile of the pallet, and the purple dashed line denotes the simulated height of the same profile. In the middle of the x axis, the simulated profile shows good conformity with the measured profile, with a 0.84 mm deviation between the experimental and numerical profiles. The warpage in the middle profile along the x axis is very small, as the heights of the profile are all below 1.29 mm and the average height of the profile is 0.49 mm.

The red circles denote the measured height on the left-hand side (*x_l_*) of the pallet and the red dotted line denotes the numerical profile. The orange triangles denote the measured height on the right-hand side (*x_r_*) of the pallet and the orange dashed line denotes the numerical profile. Numerical profiles on both the left- and right-hand sides show the convex symmetric heights of the pallet, but the experimental profiles on both sides of pallet show either positive skewness or negative skewness. The experimental deviations between the maximum and minimum heights of both profiles on the left- and right-hand sides of the pallet are 6.3 and 5.2 mm, respectively. Along the y axis, the predicted heights of the profiles also showed good conformity with the measured profile of the pallet, with a 0.92 mm deviation between the experimental and numerical profiles, as shown in Figure 4f. The middle profile (*y_c_*) along the y axis was experimentally shown to be deformed by under 1.07 mm. The experimental profiles on both sides of the pallet are characterized by positive or negative skewness. The experimental deviations between the maximum and minimum heights of both profiles, on the left-hand (*y_l_*) and right-hand (*y_r_*) sides of the pallet, are 5.9 and 4.2 mm, respectively. The numerical predictions of the profiles by Moldex3D perfectly fit the true profiles of the pallet due to the large-scale geometrical sizes of 1000 mm × 1000 mm × 130 mm.

To derive the warpage of the large-scale pallet, an injection molding process was applied, using a numerical approach. The structure of plastic pallet is composed of ribs, beams and rods, making the construction of a close-form solution inaccessible. Using Moldex3D software, one may acquire the pallet’s temperature distribution under each stage of injection molding. However, after demolding, the pallet is free to deform, and the asymmetric internal stress of the molded pallet may cause the warpage of the pallet to develop. Since the temperature in the pallet is still high and the pallet is not stiff enough to endure this, it immediately warps towards the cold side upon ejection. The hot side of the pallet has a higher temperature than the cold side; it is less stiff and experiences more cooling than the cold side as the whole pallet eventually cools down to room temperature. Hence, the hot side will deform more than the cold side during the free quench of the pallet, and this will cause the warpage to decreasingly incline toward the cold side [31,32,33]. 

### 3.3. Proposed Sequence of Valve Gate-Opening

An unbalanced filling resulting from the pre-set sequence is shown in Figure 2b,d, as well as in Figure 3b,d,f,h. The pre-set sequence shown in Table 2 is improper due to the well-maintained weld lines between the 16 gates. An essential feature of the sequential valve gate scheme is that the valve gate is actuated once the flow front spreads towards it. This arrangement may drastically reduce the number of weld lines in the injection molding parts. A newly proposed sequential valve gate scheme is shown in Table 3. The total filling time is 10.591 s, which is slightly greater than the pre-set sequence of 10.795 s.

The cooling channel temperatures within the core and cavity plates of the mold were 20 °C and 55 °C, respectively. The injection molding parameters are shown in Table 1. In the Moldex3D simulation, the flow front initially spreads from gates #6, #7, #10, and #11. Gates #2, #3, #14, and #15 are activated when the initial flow front meets the previous gates. The flow front profile is shown in Figure 5a, in which there are only three weld lines on the top surface of the pallet. The front continuously spreads to the four corners of the pallet. The gates begin filling the melt material into the cavity when the flow front meets the lateral gates, as shown in Figure 5b,c. Weld lines appear at four corner legs of the pallet, which cannot be avoided due to the two filling approaches used during the molding of the corner legs. However, this proposed filling valve gate scheme sequence eliminates a high number of weld lines on the top surface.

Figure 6a depicts the numerical flatness of the top surface of the pallet under the newly proposed sequential valve gate scheme. The top flatness of the pallet, 7.76 mm, is drastically decreased with respect to the 64.052 mm flatness by pre-setting the sequence of the valve gate-opening scheme. The profile of the top surface differs from the convex profile (Figure 4a). This reduced flatness demonstrates that the proposed sequence scheme is a successful approach, and that this flatness excellently meets the specifications of the commercial plastic pallet. Figure 6b shows the bottom surface of the pallet, with a 7.027 mm flatness. This flatness of 7.027 mm can be compared to the numerical flatness of 64.072 mm of the bottom surface of the pallet, achieved by pre-setting the valve gate system system. The proposed valve gate scheme sequence may numerically decrease due to the warpage of the rPP pallet. 

In this study, the mold temperatures of the core and cavity plates are the same, which causes a large warpage in this thin, rib-reinforced pallet due to the total thin-rib structure of the core plate. The top profiles of the pallet are, therefore, shown to be numerically in convex Figure 4a and experimentally convex in Figure 4b. Under the pre-set sequential valve gate scheme, the flatness of the pallet in Figure 7 shows that the temperature of the cavity plate of the mold has a negative proportional relationship at temperatures below 60 °C. This trend may help us to set the local mold temperature. Consequently, the simulation verifies the feasibility of using the proposed sequential valve gate system. A further experimental investigation is essential to verify the previously determined numerical flatness of the pallet. The temperatures of the core and cavity plates of the mold has to be experimentally determined via the pallet’s warpage.

## 4. Conclusions

The flatness of an rPP pallet, created using a pre-set and newly proposed sequential valve gate-opening injection molding scheme, was investigated and found to be successful. The pallet’s required specifications were achieved. The pre-set sequence of valve gate-opening produced a poor pallet product due to the weld lines present on the top surface of the pallet as the flatness neared 7 mm. The authors proposed a new sequential valve gate-opening scheme to ensure the smooth spreading of the flow front from the middle four gates. The newly proposed, eight-stage, sequential gate-opening, compared to the pre-set valve gate-opening scheme, propagates the flow front from the central gate to the four corners of the pallet, with fewer welding lines being formed between each set of four legs, located in the rib portions. The weld lines on the top surface of the pallet are, therefore, dramatically decreased. The flatness of the pallet was accurately demonstrated via experimental and numerical results. The flatness of the pallet produced by the newly proposed sequential valve gate-opening scheme is about 7 mm, which coheres with the height directional warpage achieved under the pre-set sequential scheme. By adjusting the temperature of the cooling channel within the cavity plate to 55 °C, the flatness measured by ATOS is in line with the numerical flatness of the pallet.

## Figures and Tables

**Figure 1 polymers-14-00616-f001:**
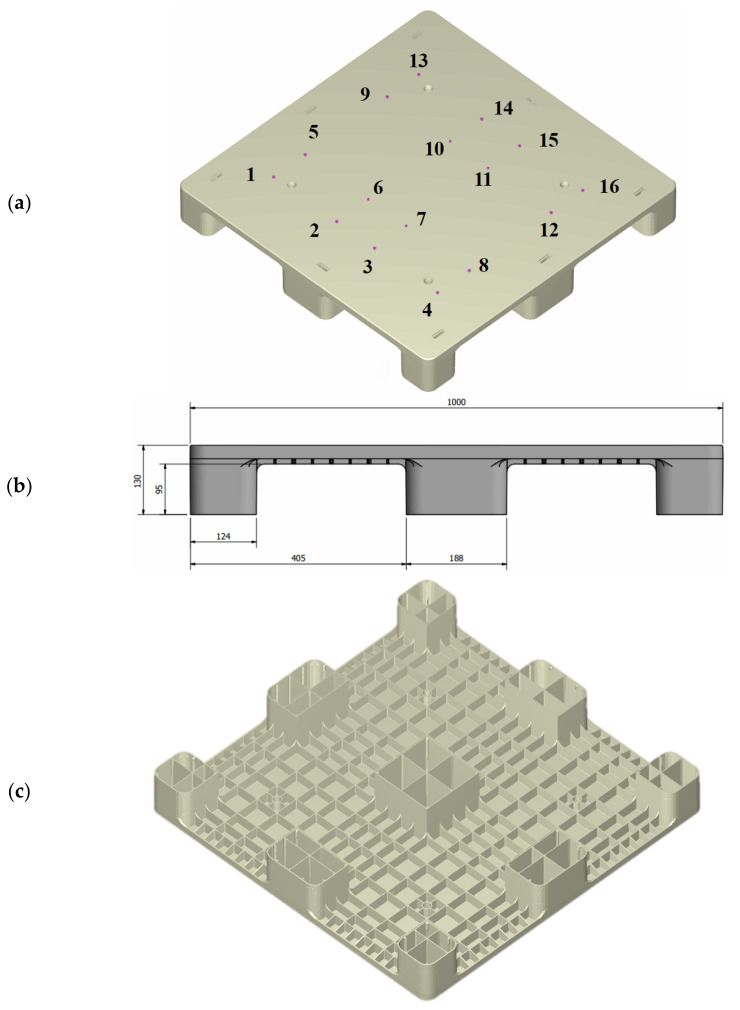
Injection-molded pallet (1000 mm × 1000 mm × 130 mm). (**a**) Sixteen filling gates in the pallet; (**b**) dimensions of the pallet (front view); (**c**) isometric view of the pallet (bottom side); (**d**) isometric view of the water flow system on the top side of the pallet; (**e**) isometric view of the baffle-cooling flow system on the bottom side of the pallet.

**Figure 2 polymers-14-00616-f002:**
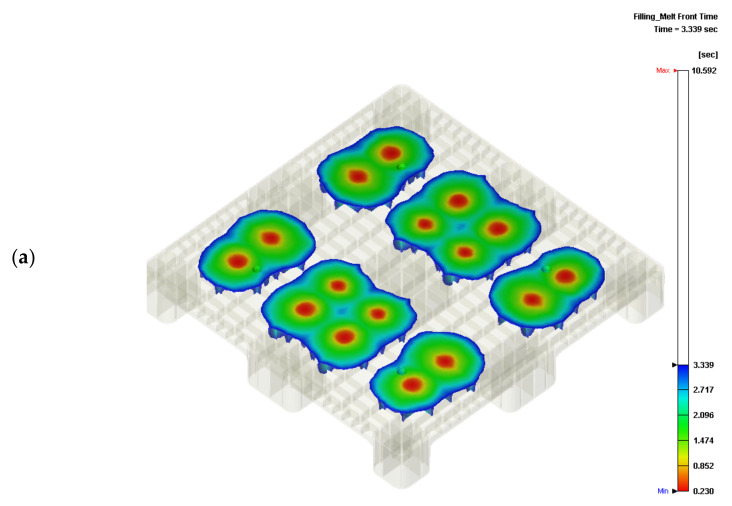
Flow fronts in short-shot testing: (**a**) 30% flow front under the concurrent valve gate-opening scheme; (**b**) 30% flow front under sequential valve gate-opening scheme; (**c**) 60% flow front under concurrent valve gat- opening scheme; (**d**) 60% flow front under sequential valve gate-opening scheme.

**Figure 3 polymers-14-00616-f003:**
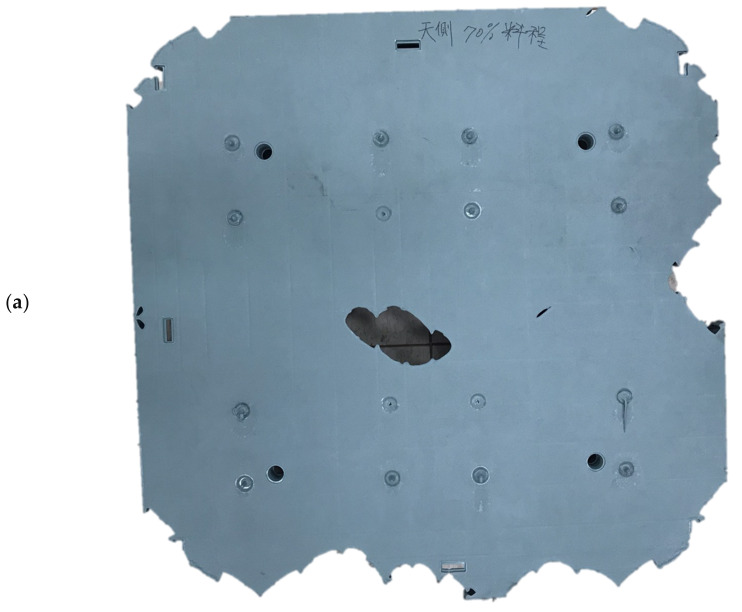
Comparisons of the experimental and numerical flow fronts in short-shot testing under the pre-set sequential valve gate-opening scheme. (**a**) Top view of the 70% short-shot pallet; (**b**) top view of the 70% numerical flow front; (**c**) bottom view of the 70% short shot pallet; (**d**) bottom view of the 70% numerical flow front; (**e**) top view of the 90% short-shot pallet; (**f**) top view of the 90% numerical flow front; (**g**) bottom view of the 90% short-shot pallet; (**h**) bottom view of the 90% numerical flow front.

**Figure 4 polymers-14-00616-f004:**
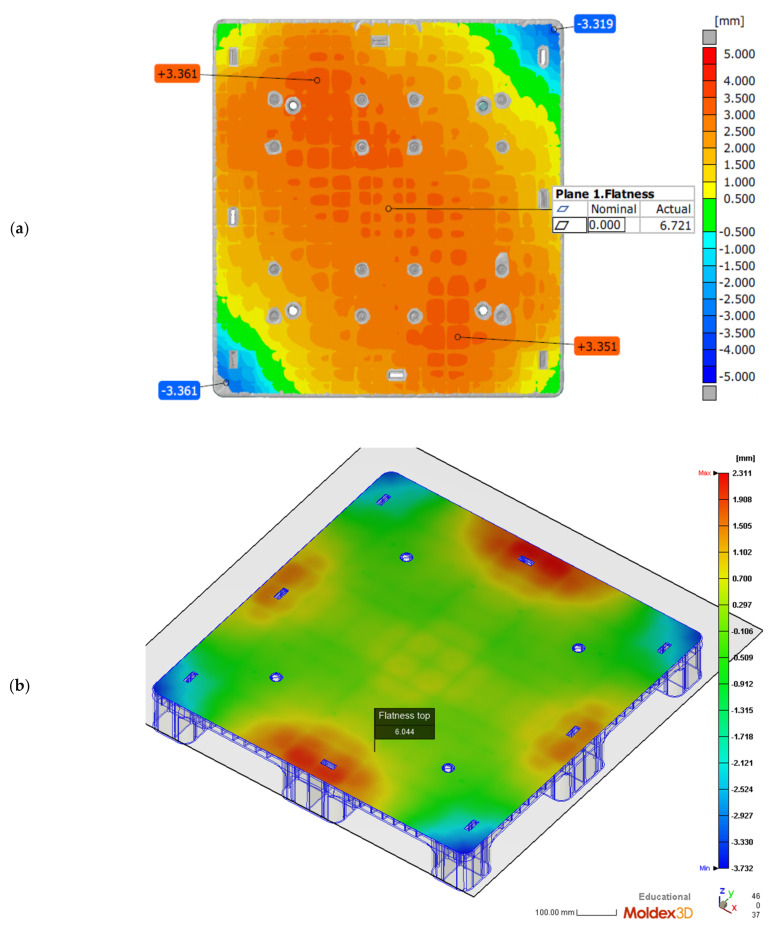
Comparisons of the numerical and experimental flatness under the pre-set sequential valve gate-opening scheme (20 °C cooling channel). (**a**) Isometric view of ATOS-measured top flatness of pallet; (**b**) Isometric view of numerical top flatness of pallet; (**c**) isometric view of ATOS-measured bottom flatness of pallet; (**d**) Isometric view of numerical bottom flatness of pallet; (**e**) x-axial top surface profiles; (**f**) y-axial top surface profiles.

**Figure 5 polymers-14-00616-f005:**
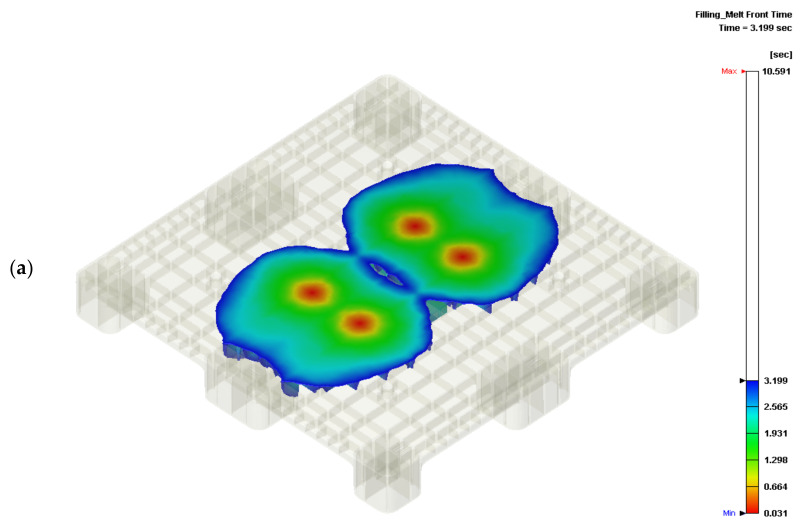
Flow front under the proposed sequential valve gate scheme: (**a**) 30% of total filling; (**b**) 60% of total filling; (**c**) 90% of total filling.

**Figure 6 polymers-14-00616-f006:**
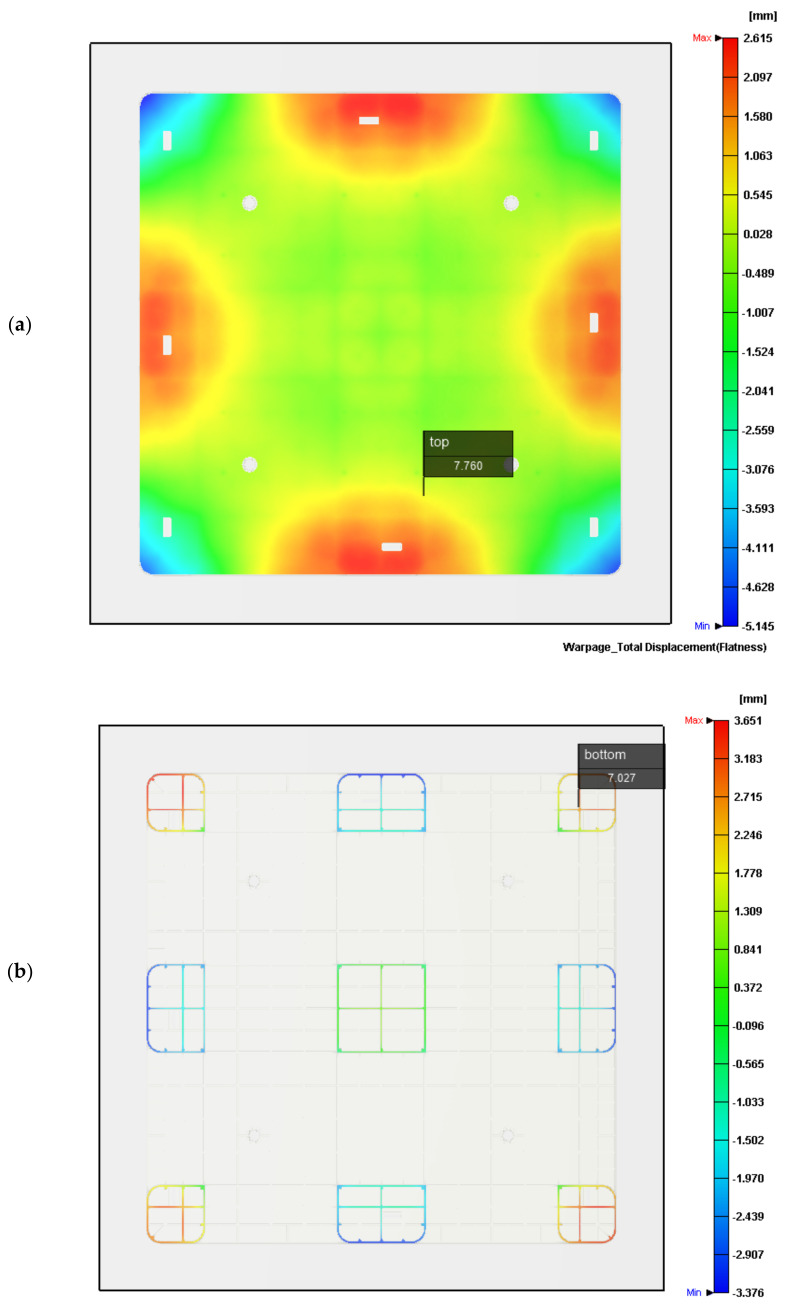
Flow front under the proposed sequential valve gate scheme: (**a**) Top flatness of pallet; (**b**) bottom flatness of pallet.

**Figure 7 polymers-14-00616-f007:**
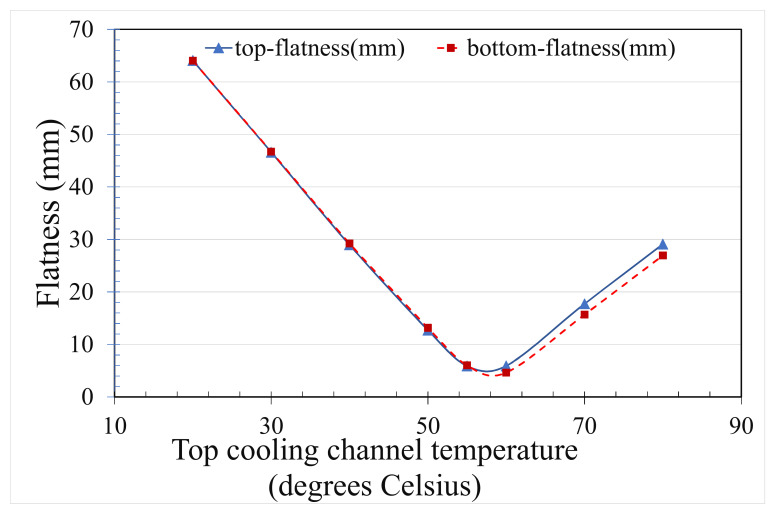
Numerical flatness with respect to the cooling channel of the cavity plate under the pre-set sequential valve gate-opening scheme (20 °C cooling channel within the core plate).

**Table 1 polymers-14-00616-t001:** Sequence of the filling gates.

Time (s)	Start	1	2	3	5	6	8
Gate (#)	#3, #5, #9, #13~16	#8, #11	#1, #10, #12	#4	#7	#6	#2

**Table 2 polymers-14-00616-t002:** Injection molding parameters of recycled polypropylene [10].

Melt temperature (°C)	250
Curing temperature (°C)	117
Mold temperature (°C)	50
Fill rate (%)	72
Filling pressure (max) [MPa]	40
Filling time (s)	9.3
Packing time (s)	3.0
Packing pressure (%)	70
Cooling time (s)	45
Mold opening time (s)	10

**Table 3 polymers-14-00616-t003:** Proposed sequence of filling valve gates.

Time (s)	Start	2.2	5.42	5.46	5.5	5.85	5.88	6.04
Gate (#)	#6, #7, #10, #11	#2, #3, #14, #15	#5	#8, #12	#9	#1	#4, #16	#13

## Data Availability

All the data generated or analyzed during this study are included in the published article.

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
