# Peer review of "Feasibility Study of the Flatness of a Plastic Injection Molded Pallet by a Newly Proposed Sequential Valve Gate System"

_polymers, 2022, doi:10.3390/polym14030616_

Round 1
Reviewer 1 Report
In the resubmitted manuscript, the authors have made much improvement to make this manuscript useful to the audience of injection molding. The results would help provide reference to the real production of polymer pallet in some aspect, for example, as stated "The flatness of the pallet achieved by the newly proposed sequential valve gate opening scheme is about 7mm, which meets the height directional warpage by the pre-set sequential scheme."
Nevertheless, the language still can be improved, since many sentences are ambiguous or inaccurate. Taking for example, the first sentence of the conclusions section, " The flatness of the rPP pallet , ... , was investigated and successful".
Moreover, the simulation setup, including materials properties, boundary conditions, step arrangement, results processing, is suggested to present in a more clear way. Especially, the reason for the sequence of the filling gates in Table 2, should be summarized.
Overall, I recommend publication of this manuscript after minor revision and language polishing.
Author Response
Dear Reviewer,
Thank you very much for your comments and suggestions. I respond as followings,
1. this revised manuscript will be arranged by MDPI for language editing to enhance readability.
2. Table 2 displayed the operational sequence of the filling gates which is provided by the incorporation. By the preliminary analysis, the authors give a good sequence as shown in Table 3 for a smaller flatness of the pallet. The simulation setup, including materials properties, boundary conditions and step arrangement are descripted behind the Figure 1. The resulting flatness of the pallet is processed in the section 3.2 Flatness.
Reviewer 2 Report
The paper entitled "Feasibility Study of the Flatness of Plastic Injection Molded Pallet by a Newly Proposed Sequential Valve Gate System" by Hsi Hsun Tsai and Yi Lin Liao compares the pallet flatness numerically by using Moldex3D solid 2020 software package and the simulation results are compared with the real injection-molded parts by using recycled polypropylene with large scale pallet .
Some major problems must be taken in account:
1) At page 8, row 419 :"The numerical and experimental flatness of the bottom surface are shown in Figure 4 (e) and (f). The numerical flatness of the bottom surface is 64.072 millimeters in Figure 4 (e), while the flatness acquired by the experiment is 6.772 mm depicted in Figure 4(f). .....This same temperature setting on both mold plates induces greater flatness in the simulation."
What is the meaning of compare simulation with experimental results if they don't fit at all? The parameters of simulation program must be adjusted in order to fit the experimental results, otherwise you cannot trust any of the conclusion of the simulation work. Next paragraph explains why is this difference, but some changes must be done in the porogram, for obtaining better results.
2) The Figures must be rearranged
3) The English language need corrections:
Author Response
1) At page 8, row 419 :"The numerical and experimental flatness of the bottom surface are shown in Figure 4 (e) and (f). The numerical flatness of the bottom surface is 64.072 millimeters in Figure 4 (e), while the flatness acquired by the experiment is 6.772 mm depicted in Figure 4(f). .....This same temperature setting on both mold plates induces greater flatness in the simulation."
What is the meaning of compare simulation with experimental results if they don't fit at all? The parameters of simulation program must be adjusted in order to fit the experimental results, otherwise you cannot trust any of the conclusion of the simulation work. Next paragraph explains why is this difference, but some changes must be done in the program, for obtaining better results.
Response:
The first analysis by the operational sequence of the filling gates as listed in Table 2 which is provided by the incorporation. By the comparison of numerical and experimental results the authors would make sure the effectiveness of the simulation. Figure 4(e) is the simulation results on the pallet profile of the bottom side which indicates a larger value on the flatness. After detail analysis again, the authors found exact flatness of the bottom side of pallet is 5.920mm. The authors thank the reviewers' reminder.
2) The Figures must be rearranged
The authors rearrange the figures of this revised manuscript.
3) The English language need corrections:
This revised manuscript will be arranged by MDPI for language editing to enhance readability.
Round 2
Reviewer 2 Report
The authors did not correct the errors and did not provide concrete explanations
Author Response
Response:
The authors mis-addressed the sentence of “The numerical flatness of the bottom surface is 64.072 millimeters in Figure 4 (e)”. We made a major revision of our manuscript as follows. Figure 4 is rearranged and added the isometric view figures for clear descriptions.
We provided the numerical flatness of the pallet by Moldex3D and fits perfectly to the ATOS measured flatness. The comparisons between the numerical and true profiles along the x and y axial directions are shown in Figure 4 (e) and (f) and indicate that numerical results meet nice to the measured results.
In the following figures, the pallet is not demolded yet. Figure (A) is an isometric view of the temperature distribution of pallet behind 45 seconds cooling where the cooling channel temperature of cavity plate of mold is 55 °C and the one of core plate is 20 °C for simulation on Moldex3D. The top side has the temperatures from 55 to 70 °C which nears the cooling channel temperature of 55 °C, so the efficiency of the cooling channel on the top side of pallet is about 0 % in Figure (B). In Figure (C) the temperatures of the legs of pallet are near about 20 °C which is the temperature of the baffle cooling channel on the bottom side of pallet. The heat from the top-side cooling channel is transferred to the bottom-side baffle cooling channel. In Figure (D), the efficiency of the bottom-side baffle cooling channel is thus about 5 %. This phenomenon, top-side temperature is greater than the bottom-side one, may help the pallet to bend to the hot side (top side) during the whole pallet cools down to room temperature after ejection stage of injection molding. In other words, the top side has a higher temperature than the bottom side. The top side is above 55 °C while the bottom side is 20 °C. The top side may have a larger shrinkage to smooth the convex profile after the demolding. The flatness of top surface of pallet is thus only 6.044 mm as shown in Figure (E).
We examine the flatness of the top surface of pallet numerically under the same temperature of all cooling channels. As depicted in Figure (F) and (H), the surface temperatures of pallet by the equal setting are lower than the one by the non-equal setting cooling temperature. The efficiencies of the top and bottom cooling channel. Comparing Figure (B) and (G), the cooling efficiencies of top-side cooling channels by the equal setting temperature are higher than the one by the non-equal setting cooling temperature. The induced flatness of top surface is 75.547 mm in Figure (J). Referred to the strength of materials, the section area of the pallet is not horizontal symmetry, the centroid of the area is not located at the middle of height of the area. During cooling down to the room temperature after demolding, the internal thermal stress occurred on the pallet section area under the equal setting of cooling temperature gives bending moment due to this asymmetric area.
The major revision is as follows,
p.15
The temperature of the cooling channel within the mold plate, including the core and cavity plates, was set to 20 °C. The injection molding samples were acquired after 20 cycles to ensure the temperature of the mold plate was in a steady state. Then the temperatures of core and cavity of mold were measured by K type thermocouple after the pallet ejected from the core plate. The measured temperatures of the previous two are higher than the cooling channel temperature since the heat within them are never transferred effectively to the cooling channel. The exact temperatures of core and cavity of mold could not be measured during the packing and cooling stages of injection molding. For fitting the numerical warpage to the molded one by the error and trial method, an empirical approach is used to adjust the temperatures of the cooling channels within the cavity and core plates of mold which induce various levels of the temperature of the two plates on the prediction of temperatures of core and cavity of mold. Hence, the entire pallet was measured by way of the ATOS scan box 5120 system, and this was then compared with the numerical results.
The difference between the highest and lowest positions of a surface profile in the height direction can be called flatness. Figure 4(a) depicts the measured flatness of top surface of molded pallet by the previous ATOS system. The entire profile of the top surface of pallet is convex and has an excellent value on flatness of 6.721 mm. By adjusting the cooling channel temperature of cavity plate of mold to 55 °C and the one of core plate to 20 °C for simulation on Moldex3D, the numerical top profile of the pallet is symmetrically convex, and the surface flatness of this top surface is 6.044 mm as shown in Figure 4 (b). This molded pallet has the feature of nine legs to support the entire top surface and the loads. The flatness of nine-leg plane of pallet is therefore important for stable supporting goods. The flatness of the nine-leg plane of pallet bottom surface is acquired by ATOS of 6.772 mm as shown in Figure 4(c). The predicted flatness of the bottom side of pallet is 5.960 mm in Figure 4(d). The deviation of 0.812 mm respect to the 130 mm in the height of pallet is relatively accurate.
(b). The comparison of experimental and numerical profiles of the molded pallet along the x axis are shown in Figure 4(e), where the purple crosses denote the measured heights every fifty millimeters along the middle (xc) of the top profile of the pallet, and the purple dashed line denotes the simulated height of the same profile. In the middle of the x axis, the simulated profile shows good conformity with the measured profile, with 0.84 millimeters of deviation between the experimental and numerical profiles. The warpage in the middle profile along the x axis is small, as the heights of the profile are all below 1.29 millimeters and the average height of the profile is 0.49 millimeters.
The red circles denote the measured height on the left-hand side (xl) of the pallet and the red dotted line denotes the numerical profile. The orange triangles denote the measured height on the right-hand side (xr) of the pallet and the orange dashed line denotes the numerical profile. Both numerical profiles on the left- and right-hand sides show the convex symmetric heights of the pallet, but the experimental profiles on both sides of pallet show either positive skewness or negative skewness. The experimental deviations between the maximum and minimum heights of both profiles on the left- and right-hand sides of the pallet are 6.3 and 5.2 millimeters, respectively. Along the y axis, the predicted heights of the profiles also showed good conformity with the measured profile of the pallet, with 0.92 millimeters in deviation between the experimental and numerical profiles, as shown in Figure 4 (f). The middle profile (yc) along the y axis was shown experimentally to be deformed less than 1.07 millimeters. The experimental profiles on both sides of pallet are characterized by positive or negative skewness. The experimental deviations between the maximum and minimum heights of both profiles on the left-hand (yl) and right-hand (yr) sides of the pallet are 5.9 and 4.2 millimeters, respectively. The numerical predictions of the profiles by Moldex3D fit perfectly to the true profiles of the pallet due to the large scale in geometrical sizes of 1000 mm x 1000 mm x 130 mm.
For deriving the warpage of the large-scale pallet in injection molding process by numerical approach is well applied. Because the structure of plastic pallet is composed of ribs, beams and rods which is inaccessible to construct a close form solution. By Moldex3D software one may acquire the temperature distribution of pallet under each stage of injection molding. However, after the demolding, the pallet is free to deform, and the asymmetric internal stress of molded pallet may cause the warpage of the pallet to develop. Since the temperature in the pallet is still high and the pallet is not stiff to endure the immediately warps toward the cold side upon ejection. The hot side of the pallet has a higher temperature than the cold side, it is less stiff and experiences more cooling than the cold side as the whole pallet eventually cools down to room temperature. The hot side thus deform more than the cold side during the free quench of the pallet, and this will cause the warpage to be less and less toward the cold side [33].

Round 3
Reviewer 2 Report
The authors made major revisions in concordance with comments
This manuscript is a resubmission of an earlier submission. The following is a list of the peer review reports and author responses from that submission.
Round 1
Reviewer 1 Report
The manuscript employs Moldex3D software to investigate the flatness of a plastic injection-molded pallet, and proposes a sequential value gate strategy. This study would be helpful in improving the quality of a large-sized pallet. However, concerning the current form of manuscript, major revision is required before making further decision. The authors should address the following issues.
- Most importantly, experimental validation of the newly proposed sequential valve gate opening scheme is a must in this study, because the reliability of simulation is poor. As stated by the authors, starting from Line 346, “the measured flatness of the bottom surface is 6.722 mm, while the flatness acquired by the simulation in Figure 4 is 64.072 mm.” The deviation is so large that the confidence on the simulation results is severely abated. Though the authors mentioned that the deviation will be reduced by adjusting the temperature of cooling channel to 55 C, the reason for this adjustment is not provided.
- The criterion or reason for the proposed sequence of valves gates in Table 3 should be strengthened. Currently, the description starting from Line 368 to line 375 is not enough. The readers expect some well-defined rules of the arrangement in the gate number, as there are as many as 16 gates.
- The authors need adding more specific details about the simulation setup, such as the material models, thermal history, boundary conditions and so on. As the deviation between simulation and experiment is very obvious, the authors should discuss the mismatch between them, and maybe try to modify the simulation setup if necessary. The definition of short shot testing should be provided.
- The organization of figures in current form does not meet the basic concise and clear requirements for academic publication. Modifications must be made to rearrange the layout.
- The language needs improvement, for example, reiteration in the abstract: “The pallet flatness is accurately demonstrated”, “The flatness of the pallet was accurately demonstrated”; Filling pressure in Table 1 is not provided; hard to understand “The numerical flatness of the top surface of the pallet is 64.052 mm, which is caused by the measured flatness of 6.721 mm”.
Reviewer 2 Report
The paper entitled "Feasibility Study of the Flatness of Plastic Injection Molded Pallet by a Newly Proposed Sequential Valve Gate System" by Hsi Hsun Tsai and co. proposed a sequential valve gate opening scheme and concluded thatits flatness is slightly larger than the ones by the pre-setting sequential scheme.
In my opinion, the paper is too long and present a short data set . The content is not very easy to read and the main achivements of the work are not very clear.
The autors concluded "The flatness of the pallet achieved by the newly proposed sequential valve gate opening scheme is about 7mm, which drastically reduces the height directional warpage than the flatness of about 65 mm by the pre-set sequential scheme. By adjusting the temperature of the cooling channel within the cavity plate to 55 °C, the measured flatness provided by ATOS is in line with the numerical flatness of the pallet."
Apart of the flatness, it is not clear what are the characteristics of the newly obtained pallet which are better compared to the already existed ones? Because in introduction the authosr said:
"The standardized pallet is designed to last for several trips and called “buy/sell” strategy. However, single-use is the simplest strategy wherein
pallets are discarded after one trip [2] which is loaded with goods into container and transported by container ships to all over the world. It was also found that the recycled plastic pallets is better than the conventional plastic pallet depending on the impact category analyzing the results per trip. The recycled plastic pallets also performed better, en40 vironmentally, than the wooden pallets [3]"
Are this newly-obtained pallets lasting more? Why is the recycled plastic (newly obtained in the paper) better than the conventional plastic pallet? Do they have better mechanical characteristics?
Reviewer 3 Report
In the manuscript, the authors studied the effect of two different filling patterns by controlling the valve gate opening sequence on the flatness of a plastic part, namely pallet. Computer simulation was conducted by using Moldex3D solid 2020 software package, and the simulation results are compared with the real injection-molded parts. However, the reviewer does not recommend an acceptance of the manuscript. Because it is now very common to use CAE software (such as Moldflow or Moldex3D) in a university or a company to solve such a problem, and to achieve a balanced flow pattern or decrease the number of weld line by using sequential valve gate are the basic goals of the simulation. Therefore, there is a lack of novelty for this manuscript. Besides, what is the purpose of section 3.1 Flow Front Comparison of Short Shot Testing? The reviewer feels that the effect of this section is just to verify the simulation accuracy of the software.